# Cancer tracking system improves timeliness of liver cancer care at a Veterans Hospital: A comparison of cohorts before and after implementation of an automated care coordination tool

**Yapei Zhang**[1,2‡], **Catherine Mezzacappa**[1‡], **Lin Shen**[3], **Amanda Ivatorov**[4], **Alexandra Petukhova-Greenstein**[5,6], **Rajni Mehta**[7], **Maria Ciarleglio**[8], **Yanhong Deng**[8], **Woody Levin**[7], **Steve Steinhardt**[7], **Donna Connery**[7], **Michael Pineau**[7], **Ifeyinwa Onyiuke**[5,7], **Caroline Taylor**[5,7], **Michal G. Rose**[1,7], **Tamar H. Taddei**[1,7] *

**1** Department of Medicine, Yale School of Medicine, New Haven, Connecticut, United States of America, **2** Department of Ophthalmology, Harvard Medical School, Boston, Massachusetts, United States of America, **3** Department of Medicine, Brigham and Women's Hospital, Boston, Massachusetts, United States of America, **4** Yale University, New Haven, Connecticut, United States of America, **5** Department of Radiology and Biomedical Imaging, Yale School of Medicine, New Haven, Connecticut, United States of America, **6** Institute of Radiology, Charité Universitätsmedizin Berlin, corporate member of Freie Universität Berlin, Humboldt-Universität, and Berlin Institute of Health, Berlin, Germany, **7** VA Connecticut Healthcare System, West Haven, Connecticut, United States of America, **8** Yale Center for Analytical Sciences, Yale School of Public Health, New Haven, Connecticut, United States of America

‡ These authors share first authorship on this work.
* tamar.taddei@va.gov

**Data Availability Statement:** De-identified data may be requested through a data use agreement

## Abstract

### Introduction

Hepatocellular carcinoma (HCC) requires complex care coordination. Patient safety may be compromised with untimely follow-up of abnormal liver imaging. This study evaluated whether an electronic case-finding and tracking system improved timeliness of HCC care.

### Methods

An electronic medical record-linked abnormal imaging identification and tracking system was implemented at a Veterans Affairs Hospital. This system reviews all liver radiology reports, generates a queue of abnormal cases for review, and maintains a queue of cancer care events with due dates and automated reminders. This is a pre-/post-intervention cohort study to evaluate whether implementation of this tracking system reduced time between HCC diagnosis and treatment and time between first liver image suspicious for HCC, specialty care, diagnosis, and treatment at a Veterans Hospital. Patients diagnosed with HCC in the 37 months before tracking system implementation were compared to patients diagnosed with HCC in the 71 months after its implementation. Linear regression was used to calculate mean change in relevant intervals of care adjusted for age, race, ethnicity, BCLC stage, and indication for first suspicious image.

with VA Connecticut Healthcare System's IRB by contacting the VA Central Institution Review Board at vacentralirb@va.gov.

**Funding:** The author(s) received no specific funding for this work.

**Competing interests:** The authors have declared that no competing interests exist.

## Results

There were 60 patients pre-intervention and 127 post-intervention. In the post-intervention group, adjusted mean time from diagnosis to treatment was 36 days shorter (p = 0.007), time from imaging to diagnosis 51 days shorter (p = 0.21), and time from imaging to treatment 87 days shorter (p = 0.05). Patients whose imaging was performed for HCC screening had the greatest improvement in time from diagnosis to treatment (63 days, p = 0.02) and from first suspicious image to treatment (179 days, p = 0.03). The post-intervention group also had a greater proportion of HCC diagnosed at earlier BCLC stages (p<0.03).

## Conclusions

The tracking system improved timeliness of HCC diagnosis and treatment and may be useful for improving HCC care delivery, including in health systems already implementing HCC screening.

## Author summary

Liver cancer typically occurs in people with advanced scarring of the liver, a condition called cirrhosis. Survival in liver cancer depends on both the stage of cancer and the severity of pre-existing liver disease. Treatment requires coordination between multiple subspecialties: Hepatology, Oncology, Interventional Radiology, and Surgery. The most common causes of liver cirrhosis in the United States are hepatitis C virus and alcohol use, and patients with cirrhosis often face difficulties accessing timely care due to coexisting mental health conditions and health systems barriers. We implemented a web-based tracking system at the Veterans Affairs (VA) Connecticut Healthcare System that 1) automatically generates queues of abnormal imaging suspicious for liver cancer using diagnostic codes and natural language processing, and 2) maintains queues of tasks for follow up diagnostic testing and cancer treatment. We found implementation of this tracking system reduced time from first image suspicious for liver cancer to first cancer treatment. Advances in diagnostic technology and treatment options for liver cancer will benefit patients the most if patients are connected to appropriate specialty treatment in a timely manner, and our study demonstrates the potential of this practical intervention to systematically address gaps in care in an at-risk population.

## Introduction

Liver cancer is the sixth most commonly diagnosed cancer and the fourth leading cause of cancer mortality worldwide, contributing to 782,000 deaths annually [1]. Hepatocellular carcinoma (HCC) accounts for 75% of liver cancer cases [2] and is a complex disease that usually occurs in the setting of chronic liver disease. In the United States, HCC incidence rates have tripled over the last thirty years [3], and among all cancers in the US, HCC incidence is growing most rapidly [4]. The prognosis of HCC is determined not only by stage at diagnosis, but also by the etiology and severity of the underlying liver disease [5,6]. Its management requires timely and extensive coordination and communication between different specialists and multiple follow-up visits. As a result, patients with HCC are susceptible to treatment delays, and

many do not receive any cancer treatment [7]. A study at a large urban hospital showed that in 38.5% of patients with HCC, the time from presentation to diagnosis exceeded 90 days [8].

The VA Healthcare System is the largest single care provider to patients with chronic liver disease in the United States [9]. As such, the VA coordinates the care of many patients with HCC. The VA patient population is especially susceptible to treatment delays, as it is enriched with patients suffering from mental illness, homelessness, and alcohol and substance use disorders [10].

To address treatment delays at VA Connecticut Healthcare System (VACHS), we implemented a web-based, electronic medical record (EMR)-linked, tracking system to coordinate liver cancer care. The system uses diagnostic codes and a natural language processor to review all abdominal imaging performed at VACHS daily to generate a list of patients with imaging suspicious for HCC. Using pre-defined criteria, nurse coordinators bring selected cases to a multidisciplinary liver tumor board (MDTB) attended by hepatology, radiology, surgery, interventional radiology, pathology and medical oncology, which was established in February 2009. An individualized work-up and treatment plan is generated and documented in the EMR which ensures communication between all treating specialties and the patients' primary care provider. The nurse coordinators oversee the plan's implementation and communicate with patients as needed. The electronic tracking platform also serves as a reminder system to ensure that follow-up studies and treatment occurs and to generate timeliness and other reports. We previously demonstrated that its implementation for chest imaging was associated with a mean reduction of 25 days between the first suspicion of lung cancer and the initiation of treatment [11]. We hypothesized that this tracking system would similarly improve the timeliness of HCC care.

## Methods

### Study design

We conducted a pre-/post-intervention study to evaluate whether implementation of a liver cancer tracking system reduced the time between liver imaging suspicious for HCC and first specialty care appointment, HCC diagnosis, and treatment of HCC. The tracking system includes automated review of all relevant radiology reports using designated radiology codes and a natural language processor, which flags findings possibly concerning for hepatocellular carcinoma. Flagged lesions are placed in a queue that is triaged daily by a cancer care coordinator. Upon review, if a lesion requires further action, the care coordinator can assign follow-up tasks with due dates for subsequent steps in cancer care. Tasks include clinical care events like referral for specialist appointments, tumor board discussion, or surveillance imaging and are maintained in a task queue in the tracking system. Patients referred to MDTB may be recommended for no further testing, surveillance of the suspicious lesion with serial imaging, diagnostic testing (dedicated imaging protocoled for diagnosis of HCC or tissue biopsy), or treatment. For patients diagnosed with HCC, the tracking system generates alerts for upcoming treatment events as well as incomplete or missed events. This tracking system was implemented on February 1st, 2010. We compared the time from first imaging suspicious for HCC to diagnosis and from HCC diagnosis to first treatment in patients diagnosed with HCC in the 37 months before tracking system implementation (pre-intervention, January 1st, 2007 through January 31st, 2009) to patients diagnosed with HCC in the 71 months after tracking system implementation (post-intervention, January 1st, 2011 through December 31st, 2015). We began measuring the post-intervention period after an 11-month grace period for care coordinators to become trained and to incorporate the tracking system into their workflow.

## Study subjects

Eligible subjects were identified through the tumor registry and included all patients with confirmed HCC diagnosed by imaging or biopsy treated at VACHS between the years 2007 and 2015. For patients with multiple liver tumors, only the workup and management of the first tumor was assessed for this study. We excluded patients who were diagnosed with HCC outside VACHS and those who were diagnosed at VACHS but received their care elsewhere. The study was approved by the VACHS IRB.

## Variables

Variables collected on each subject included demographics, Barcelona Clinic Liver Cancer (BCLC) stage at time of HCC diagnosis, the type of initial imaging that led to the diagnosis (ultrasound [US], computed tomography [CT], or magnetic resonance imaging [MRI]), and whether the initial imaging study suspicious for HCC was conducted for HCC screening, for evaluation of patient-reported symptoms (e.g. right upper quadrant pain) or provider-identified signs of HCC including biochemical abnormalities (abnormal liver chemistry or alpha-fetoprotein), or was incidentally found on imaging performed for other purposes (e.g. lung cancer screening). Our primary outcome was time from diagnosis to initiation of first HCC-targeted treatment. Secondary outcomes included time from initial suspicious liver image to first specialty care appointment (overall and stratified by clinic type: hepatology vs. medical or surgical oncology), to HCC diagnosis, and to treatment initiation. For patients who received best supportive care, their first treatment date was defined as the date when the decision was made by the patient and provider to opt for best supportive care.

Date of diagnosis was determined by date of tissue diagnosis, by date of a definitive radiology report of HCC from multi-phase contrast-enhanced CT or MRI, or by MDTB consensus. For patients without tissue diagnosis, diagnosis by imaging was used. If the imaging report stated the lesion was "compatible with HCC," "consistent with HCC," "confirms suspicion of HCC," or stated that the lesion met Li-RADS 4–5 [12] or OPTN class 5 criteria [13], then the image was considered diagnostic of HCC. If the report stated that the liver lesion was "suspicious for HCC" and the patient was then referred for treatment based on this report, then the image was considered diagnostic of HCC. If further diagnostic study or MDTB review was conducted as the next step to confirm the diagnosis, then the image was considered non-diagnostic of HCC. Diagnosis by MDTB consensus occurred when diagnostic reports did not meet above criteria. The date of first suspicious image was defined as the date of the first liver image suspicious for HCC. If after the first suspicious image the patient was placed under surveillance with serial imaging, the number of surveillance imaging studies and the date of the study that demonstrated tumor progression warranting either confirmatory imaging, biopsy, or treatment as the next step in management were also ascertained.

## Data analysis

Demographic and clinical characteristics were compared across the pre-intervention and post-intervention groups using chi-squared tests for categorical variables and two-sample t-tests and Wilcoxon rank sum tests for continuous variables. Unadjusted time from HCC diagnosis to treatment across intervention periods was compared using generalized linear models with patient demographic characteristics, BCLC stage at diagnosis, and indication for first suspicious imaging as predictors with interaction terms between the predictor of interest and period (pre- versus post-intervention) for each model. Multiple linear regression was used to calculate mean change in HCC care intervals adjusted for patient age, race, Hispanic ethnicity, state of residence (defined as CT, regional states [NY, MA, RI, NJ], and other states), BCLC stage at

diagnosis, and indication for first suspicious imaging (screening, symptomatic, or incidental). All statistical analyses were performed using SAS 9.4 (SAS Institute, NC, USA).

### Ethics statement

This study protocol was reviewed and approved by the VA Connecticut Healthcare System Human Studies Subcommittee, approval number 1582913. The study was exempted from requiring written informed consent.

## Results

Over the 5-year intervention period, a total of 570 abnormal images were designated by the care coordinator as needing further action, ranging from 92 to 157 per year. Of the 570 lesions flagged, 218 (38.2%) were malignant. Of the 218 malignancies, 127 (58.3%) were a first diagnosis of primary HCC at VACHS and were included in the post-intervention analytic sample. The pre-intervention cohort consisted of 60 patients with a first incidence of HCC diagnosed at VACHS before tracking system implementation.

There was no significant difference in the age, sex, race, ethnicity, or primary state of residence distributions between the two groups. The pre-intervention cohort had a lower proportion of patients with BCLC stage A (29% v. 46%) or B (12% v. 20%) HCC and a higher proportion of patients with BCLC stage C (29% v. 14%) or stage D (21% vs. 12%) HCC at time of diagnosis compared to the post-intervention cohort (p = 0.029) (**Table 1**). There were no significant differences in the indication for first suspicious imaging study (screening, symptomatic, or incidental), in the distribution of initial imaging modality (US, CT, or MRI), or in the proportion of patients placed under surveillance with serial imaging (**Table 2**). Patients

**Table 1. Demographic and clinical characteristics of patients diagnosed with HCC pre- and post-intervention.**

| Characteristics | Pre-intervention N = 60 | Post-intervention N = 127 | P |
|---|---|---|---|
| **Age**, Mean (SD), years | 63.9 (10.0) | 64.2 (8.2) | 0.78 |
| **Male**, No. (%) | 60 (100) | 127 (99) | 0.49 |
| **Race**, No. (%) | | | 0.09 |
| White | 39 (65) | 103 (81) | |
| Black or African American | 14 (23) | 14 (11) | |
| Native Hawaiian or Other Pacific Islander | 1 (2) | 1 (1) | |
| American Indian or Alaska Native | 0 (0) | 2 (2) | |
| Other/Unknown | 6 (10) | 7 (6) | |
| **Ethnicity**, No. (%) | | | 0.95 |
| Hispanic or Latino | 5 (8) | 10 (8) | |
| Not Hispanic or Latino | 54 (90) | 114 (90) | |
| Unknown | 1 (2) | 3 (2) | |
| **State of Residence**, No. (%) | | | 0.45 |
| Connecticut | 43 (72) | 90 (71) | |
| Regional States (NY/MA/RI/NJ) | 10 (17) | 28 (22) | |
| Other/Unknown | 7 (12) | 9 (7) | |
| **BCLC Stage at Time of HCC Diagnosis**, No. (%) | | | 0.03 |
| 0 | 5 (9) | 10 (8) | |
| A | 17 (29) | 59 (46) | |
| B | 7 (12) | 25 (20) | |
| C | 17 (29) | 18 (14) | |
| D | 12 (21) | 15 (12) | |

**Table 2. First suspicious imaging and surveillance imaging in patients diagnosed with HCC.**

|  | Pre-intervention N = 60 | Post-intervention N = 127 | P |
|---|---|---|---|
| **Indication for first suspicious image**, No. (%) |  |  | 0.95 |
| Screening | 26 (43) | 57 (45) |  |
| Symptomatic | 22 (37) | 46 (37) |  |
| Incidental | 12 (20) | 23 (18) |  |
| **Modality of first suspicious image**, No. (%) |  |  | 0.10 |
| US | 26 (43) | 65 (51) |  |
| CT | 29 (48) | 42 (33) |  |
| MRI | 5 (8) | 20 (16) |  |
| **Placed under surveillance**, No. (%) | 11 (18) | 25 (20) | 0.83 |
| **Number surveillance images**, Median (IQR) | 2 (1) | 2 (2) | 0.26[a] |
| **Duration of surveillance**, Median (IQR), days | 246 (209) | 209 (208) | 0.59[a] |

a Calculated using Wilcoxon two-sample test for non-parametric data.

placed under surveillance had a median of 2 total surveillance imaging studies after first suspicious image both pre- and post-intervention and there was no significant difference in duration of surveillance pre- and post-intervention.

In the unadjusted analysis, the primary outcome, mean time from HCC diagnosis to initiation of first treatment, was reduced from 86.0 to 58.2 days after implementation of the tracking system (p = 0.03). There were significant interactions between intervention period and White race, non-Hispanic ethnicity, residence in Connecticut, imaging performed for screening, and not being enrolled in surveillance with serial imaging. There was a trend toward shorter mean intervals from diagnosis to initiation of first treatment among patients with more advanced HCC at diagnosis (BCLC stage C or D) compared to patients with earlier stage HCC (BCLC stage 0, A, or B) in both time periods. Change in mean time from HCC diagnosis to first treatment post-intervention was significantly different only in patients with BCLC stage A HCC (58.5 days v. 119.2 days), while all other differences by stage at diagnosis were not statistically significant. (**Table 3**)

Time from HCC diagnosis to treatment decreased by an average of 36 days post intervention (p = 0.007) when adjusting for age, race, ethnicity, state of residence, BCLC stage at diagnosis, and indication for first suspicious image (**Fig 1**).

In adjusted analyses, the post-intervention group also had a non-statistically significant mean reduction of 51 days in time from first suspicious imaging to diagnosis (p = 0.21) and a mean reduction of 87 days in time from first suspicious image to treatment initiation (p = 0.05) compared to the pre-intervention group. (**Table 4**)

Sensitivity analyses excluding patients who were placed under surveillance with serial imaging yielded similar results to the total sample. Adjusted mean time from HCC diagnosis to initiation of first treatment was reduced by 39 days (p = 0.008) with a non-statistically significant mean decrease in time from first suspicious imaging to diagnosis of 28 days (p = 0.08) and a significant decrease in time from first suspicious imaging to first treatment of 69 days (p = 0.003). (**Table 4**)

Stratified by indication for first suspicious image, patients whose imaging was performed as part of HCC screening had a significant decrease in time from diagnosis to treatment initiation (63 days, p = 0.015) while those whose first suspicious liver image was an incidental finding or to evaluate symptoms of possible HCC had non-statistically significant decreases in time from diagnosis to treatment (40 days with p = 0.11 and 29 days with p = 0.14, respectively).

**Table 3. Time from HCC diagnosis to treatment by patient characteristics.**

| | Pre-intervention Mean (95% CI) | Post-intervention Mean (95% CI) | P[a] |
|---|---|---|---|
| **Total sample** | 86.0 (65.7–106.2) | 58.2 (44.7–71.7) | 0.03 |
| **Race** | | | |
| White | 93.3 (67.8–118.9) | 57.2 (42.0–72.4) | 0.02 |
| Black or African American | 85.0 (43.1–126.9) | 59.6 (19.2–100.0) | 0.39 |
| Native Hawaiian or Pacific Islander | - | 49.0 (-102.2–100.2) | - |
| American Indian or Alaska Native | - | 84.5 (-22.4–191.4) | - |
| Other/Unknown | 45.7 (-16.1–107.4) | 63.4 (6.3–120.6) | 0.68 |
| **Ethnicity** | | | |
| Hispanic or Latino | 90.3 (15.2–165.3) | 78.0 (28.0–128.0) | 0.79 |
| Not Hispanic or Latino | 85.7 (64.5–106.9) | 57.8 (43.5–72.0) | 0.03 |
| Unknown | - | 14.0 (-72.7–100.7) | - |
| **State of residence** | | | |
| Connecticut | 93.9 (70.0–117.9) | 55.8 (39.7–71.8) | 0.0097 |
| Regional States (NY/MA/RI/NJ) | 89.8 (36.8–142.7) | 67.3 (38.5–96.1) | 0.67 |
| Other/Unknown | 37.7 (-18.8–94.3) | 54.1 (4.22–104.0) | 0.46 |
| **BCLC Stage at Time of HCC Diagnosis** | | | |
| 0 | 81.8 (15.6–148.0) | 62.5 (15.7–109.3) | 0.64 |
| A | 119.2 (83.3–155.1) | 58.5 (39.1–77.9) | 0.0038 |
| B | 119.0 (63.1–174.9) | 73.3 (43.0–103.5) | 0.16 |
| C | 47.7 (9.5–85.9) | 35.4 (-1.6–72.4) | 0.65 |
| D | 66.2 (19.4–113.0) | 54.3 (16.1–92.5) | 0.70 |
| **Indication for first suspicious image** | | | |
| Screening | 113.7 (82.7–144.6) | 67.5 (47.6–87.3) | 0.014 |
| Symptomatic | 59.2 (26.0–92.4) | 51.7 (29.3–87.3) | 0.71 |
| Incidental | 77.1 (32.3–121.9) | 48.9 (17.2–80.6) | 0.31 |
| **Placed under surveillance** | | | |
| No | 85.9 (63.5–108.4) | 57.3 (42.1–72.5) | 0.039 |
| Yes | 86.6 (36.4–136.7) | 61.5 (31.4–91.6) | 0.40 |

a All p-values are for the interaction term between patient characteristic and period (pre- versus post-intervention) except for the first row reflecting the total sample, for which the p-value is for the comparison of pre- versus post-intervention alone.

Similarly, patients whose imaging was performed as part of HCC screening had a significant decrease in time from first suspicious image to treatment (179 days, p = 0.03), while the decreases among those whose first suspicious image was incidental (68 days) or was performed

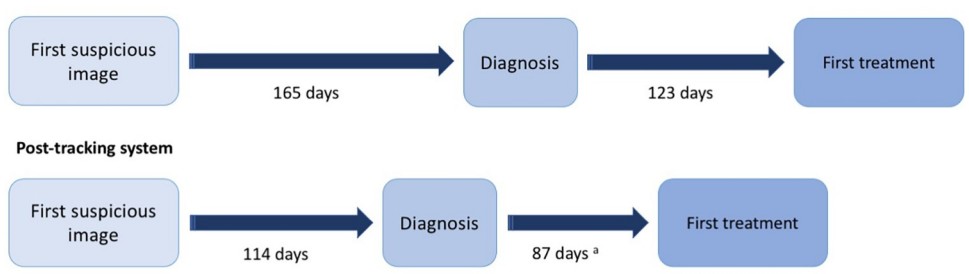

**Fig 1. Duration of time elapsed (in days) from first image suspicious for liver cancer to diagnosis and first treatment pre- and post-implementation of the tracking system.**

**Table 4. Adjusted<sup>a</sup> mean decrease in days elapsed in the cascade of HCC care post-intervention.**

|  | Decrease (in days) for all patients Mean (95% CI) | P | Decrease (in days) excluding patients placed under surveillance Mean (95% CI) | P |
|---|---|---|---|---|
| Diagnosis to first HCC treatment[b] | 36 (10–62) | 0.007 | 39 (11–68) | 0.008 |
| Screening | 63 (13–114) | 0.015 | 92 (24–161) | 0.009 |
| Symptomatic | 29 (-10–68) | 0.14 | 32 (-8–73) | 0.11 |
| Incidental | 40 (-10–90) | 0.11 | 36 (-37–109) | 0.31 |
| First suspicious image to. . . |  |  |  |  |
| First specialist appointment | 16 (-54–87) | 0.65 | 16 (-14–47) | 0.30 |
| First hepatology appointment | 13 (-60–87) | 0.72 | 19 (-14–52) | 0.25 |
| First oncology appointment | 45 (-24–114) | 0.20 | 55 (8–101) | 0.02 |
| First MDTB[c] discussion | 149 (42–257) | 0.007 | 112 (65–159) | <0.0001 |
| HCC diagnosis | 51 (-29–132) | 0.21 | 28 (-3–59) | 0.08 |
| First HCC treatment[b] | 87 (0–175) | 0.05 | 69 (24–114) | 0.003 |
| Screening | 179 (23–336) | 0.03 | 108 (26–189) | 0.01 |
| Symptomatic | 10 (-58–79) | 0.77 | 36 (-25–97) | 0.25 |
| Incidental | 68 (-133–270) | 0.49 | 168 (-9–344) | 0.06 |

a Adjusted for BCLC stage at diagnosis, age, race, Hispanic ethnicity, state of residence, and indication for first abnormal imaging.

b Sub-analyses stratified by indication for first suspicious imaging adjusted for BCLC stage at diagnosis, age, race, Hispanic ethnicity, and state of residence.

c MDTB = Multidisciplinary tumor board

for symptoms (10 days) were not statistically significant. Excluding patients placed under surveillance with serial imaging, the time from first suspicious image to treatment decreased by 168 days (p = 0.06) among those whose liver lesions were identified incidentally. (**Table 4**)

No significant difference was observed between the pre- and post-intervention cohorts in imaging-to-first-specialist-appointment time (p = 0.65). Time from first suspicious image to first MDTB consultation decreased by 149 days in the post-intervention group (p = 0.007). (**Table 4**)

## Discussion

In this retrospective pre-/post-intervention study, we found that implementation of an EMR-linked cancer identification and tracking system supported by care coordinators and a MDTB was associated with improved timeliness of HCC care at a Veterans Affairs hospital in patients diagnosed with HCC identified by screening and patients diagnosed with early-stage HCC. There were no system-level changes in access to specialists such as hepatologists, liver surgeons, interventional radiologists, or oncologists during our study period to explain the improved timeliness of care.

A combination of both patient and health system barriers may affect timeliness of HCC care. Many veterans with HCC also struggle with psychosocial challenges including psychiatric disease, alcohol and substance use disorders, and homelessness [14], which can complicate planning and care delivery and result in treatment delays [15]. Health system barriers include lack of resources to coordinate complex, multidisciplinary care and lack of follow up of incidentally discovered liver cancers. The importance of cancer care coordinators has been recognized for other malignancies [11,16–19] but has not been demonstrated previously for HCC. Our study demonstrates that a cancer care coordination system comprised of automated flagging of imaging suspicious for HCC, nurse navigators, and MDTB can have a significant impact on timeliness of care in addition to clinical HCC screening guidelines.

We attempted to minimize differences attributable to patient barriers by comparing patients from the same VA medical center and by including basic demographic characteristics as covariates in adjusted models. The post-intervention group consisted of a higher proportion of patients who self-identified as White. We observed a reduction in time from HCC diagnosis to treatment for White patients and non-Hispanic patients after implementation of the tracking system but the differences observed in Black and Hispanic patients were not statistically significant, likely due to limited sample sizes. A VA study by Zullig et al. on the timeliness of care for non-small cell lung cancer did not demonstrate a significant relationship between race and time from diagnosis to treatment [20]. However, another VA study by Merkow et al. on treatment wait times for colorectal cancer found that Black patients had longer median time to first treatment compared to White patients [21]. The results of our single center study should be interpreted with caution within the context of larger efforts to better understand the role of race in cancer care.

We observed a greater proportion of patients diagnosed with early BCLC stage HCC in the post-intervention period compared to the pre-intervention period. The etiology of this favorable stage migration is likely multifactorial and in part attributable to an increased uptake of liver cancer screening and the availability of highly effective hepatitis C treatment during the study period, as well as our cancer identification and tracking system that ensured that patients with early-stage cancers were referred for treatment. Although we observed a trend toward shorter time from diagnosis to first treatment for every BCLC stage, the improvement was statistically significant only for BCLC stage A disease. At the time of diagnosis and staging, appropriate patients with localized tumors and adequate functional reserve are referred directly for tumor ablation or embolization by Interventional Radiology. Improvements in timely referral for embolization or ablation of early-stage HCC likely drive the observed decrease in time to first treatment even where time to first visit with a hepatologist, oncologist, or surgeon was not significantly shorter.

Our study has several limitations. Due to the small sample size, several comparisons performed within subgroups failed to meet statistical significance but may represent clinically meaningful trends. The adjusted linear regression models presented in Table 4 stratified by indication for first suspicious image were performed on small sub-sample sizes and should be interpreted with this in mind. While this was a retrospective study, the data utilized were prospectively collected as part of the medical record. As a single institution study with primarily male and White patients, results from this study may not be generalizable to all healthcare facilities or patient populations. Finally, some patients with small or borderline-malignant lesions may have been initially managed by surveillance ("watchful waiting") with serial imaging, which could prolong the time from first suspicious imaging to various time points examined in this study. Time from diagnosis to treatment is independent of "watchful waiting" during a surveillance period and was thus designated as the primary outcome of this study. We also addressed this potential bias by performing sensitivity analyses excluding those who were placed under surveillance.

Since its first implementation at VACHS in 2010, this cancer tracking system has been implemented at 20 VA medical centers across the United States. Furthermore, use of this tracking system, developed initially for lung and liver cancers, has been extended to the care management of other cancers. Therefore, we hope to study the effects of this tracking system on health care coordination and delivery for other cancers and chronic illnesses on a much larger scale.

## Conclusion

Results suggest that the liver cancer tracking system implemented at VACHS in 2010 has improved the timeliness of HCC diagnosis and treatment at this institution. The complexities of care addressed by our tracking system are not unique to liver cancer, and this type of tool may improve cancer care coordination and delivery across diseases and health systems.

## Author Contributions

**Conceptualization:** Yapei Zhang, Catherine Mezzacappa, Woody Levin, Steve Steinhardt, Caroline Taylor, Michal G. Rose, Tamar H. Taddei.

**Data curation:** Yapei Zhang, Catherine Mezzacappa, Lin Shen, Amanda Ivatorov, Alexandra Petukhova-Greenstein, Rajni Mehta, Donna Connery.

**Formal analysis:** Yapei Zhang, Catherine Mezzacappa, Lin Shen, Alexandra Petukhova-Greenstein, Maria Ciarleglio, Yanhong Deng.

**Methodology:** Catherine Mezzacappa, Michael Pineau, Ifeyinwa Onyiuke, Michal G. Rose, Tamar H. Taddei.

**Project administration:** Rajni Mehta.

**Software:** Woody Levin, Steve Steinhardt.

**Supervision:** Tamar H. Taddei.

**Writing – original draft:** Yapei Zhang, Catherine Mezzacappa, Lin Shen, Tamar H. Taddei.

**Writing – review & editing:** Catherine Mezzacappa, Lin Shen, Amanda Ivatorov, Alexandra Petukhova-Greenstein, Rajni Mehta, Maria Ciarleglio, Yanhong Deng, Woody Levin, Steve Steinhardt, Donna Connery, Michael Pineau, Ifeyinwa Onyiuke, Caroline Taylor, Michal G. Rose, Tamar H. Taddei.

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
