## [Decision Letter · Decision Letter 0]

1 May 2022

PDIG-D-22-00045

Cancer tracking system improves timeliness of liver cancer care at a Veterans Hospital: a comparison of cohorts before and after implementation of an automated care coordination tool

PLOS Digital Health

Dear Dr. Taddei,

Thank you for submitting your manuscript to PLOS Digital Health. After careful consideration, we feel that it has merit but does not fully meet PLOS Digital Health's publication criteria as it currently stands. Therefore, we invite you to submit a revised version of the manuscript that addresses the points raised during the review process.

Please submit your revised manuscript by . If you will need more time than this to complete your revisions, please reply to this message or contact the journal office at digitalhealth@plos.org. Please include the following items when submitting your revised manuscript:

We look forward to receiving your revised manuscript.

Kind regards,

Shlomo Berkovsky

Section Editor

PLOS Digital Health

Journal Requirements:

1. Please provide separate figure files in .tif or .eps format and ensure that all files are under our size limit of 10MB.

For more information about how to convert your figure files please see our guidelines: https://journals.plos.org/digitalhealth/s/figures

Additional Editor Comments (if provided):

Reviewers' comments:

Reviewer's Responses to Questions

**Comments to the Author**

1. Does this manuscript meet PLOS Digital Health’s publication criteria? Is the manuscript technically sound, and do the data support the conclusions? The manuscript must describe methodologically and ethically rigorous research with conclusions that are appropriately drawn based on the data presented.

Reviewer #1: Yes

2. Has the statistical analysis been performed appropriately and rigorously?

Reviewer #1: Yes

3. Have the authors made all data underlying the findings in their manuscript fully available (please refer to the Data Availability Statement at the start of the manuscript PDF file)?

Reviewer #1: Yes

4. Is the manuscript presented in an intelligible fashion and written in standard English?

Reviewer #1: Yes

5. Review Comments to the Author

Reviewer #1: Zhang et al in their manuscript entitled "Cancer tracking system improves timeliness of liver cancer care at a Veterans Hospital: a comparison of cohorts before and after implementation of an automated care coordination tool" evaluate a web-based, EMR linked tracking system to coordinate liver cancer care. This work builds on prior efforts in lung cancer screening. This is a well-designed study of an important intervention. The limitations are noted. Was the tumor board up and running in the pre implentation period? Please discuss why the time to see a specialist was not different between groups but the time from diagnosis to treatment was less?

6. PLOS authors have the option to publish the peer review history of their article (what does this mean?). If published, this will include your full peer review and any attached files.

**Do you want your identity to be public for this peer review?** For information about this choice, including consent withdrawal, please see our Privacy Policy.

Reviewer #1: Yes: David Bentrem

---

## [Editor Report · Decision Letter 1]

26 Jun 2022

Cancer tracking system improves timeliness of liver cancer care at a Veterans Hospital: a comparison of cohorts before and after implementation of an automated care coordination tool

PDIG-D-22-00045R1

Dear Dr. Taddei,

We are pleased to inform you that your manuscript 'Cancer tracking system improves timeliness of liver cancer care at a Veterans Hospital: a comparison of cohorts before and after implementation of an automated care coordination tool' has been provisionally accepted for publication in PLOS Digital Health.

Best regards,

Shlomo Berkovsky

Section Editor

PLOS Digital Health